# Unleash Multifunctional Role of miRNA Biogenesis Gene Variants (*XPO5*rs34324334* and *RAN*rs14035*) with Susceptibility to Hepatocellular Carcinoma

**DOI:** 10.3390/jpm13060959

**Published:** 2023-06-06

**Authors:** Mohamed I. Elsalahaty, Afrah F. Salama, Thoria Diab, Medhat Ghazy, Eman Toraih, Rami M. Elshazli

**Affiliations:** 1Biochemistry Division, Department of Chemistry, Faculty of Science, Tanta University, Tanta 31527, Egypt; 2Department of Internal Medicine, Faculty of Medicine, Tanta University, Tanta 31527, Egypt; 3Endocrine and Oncology Division, Department of Surgery, School of Medicine, Tulane University, New Orleans, LA 70112, USA; 4Department of Histology and Cell Biology, Genetics Unit, Faculty of Medicine, Suez Canal University, Ismailia 41522, Egypt; 5Biochemistry and Molecular Genetics Unit, Department of Basic Sciences, Faculty of Physical Therapy, Horus University—Egypt, New Damietta 34517, Egypt

**Keywords:** *XPO5*rs34324334*, *RAN*rs14035*, gene vriants and HCC

## Abstract

Numerous reports have explored the roles of different genetic variants in miRNA biogenesis mechanisms and the progression of various types of carcinomas. The goal of this study is to explore the association between *XPO5*rs34324334* and *RAN*rs14035* gene variants and susceptibility to hepatocellular carcinoma (HCC). In a cohort of 234 participants (107 HCC patients and 127 unrelated cancer-free controls) from the same geographic region, we characterized allelic discrimination using PCR-RFLP and performed subgroup analysis and multivariate regression. We found that the frequency of the *XPO5*rs34324334* (A) variant was correlated with elevated risk of HCC under allelic (OR = 10.09, *p*-value < 0.001), recessive (OR = 24.1, *p*-value < 0.001), and dominant (OR = 10.1, *p*-value < 0.001) models. A/A genotype was associated with hepatitis C cirrhosis (*p*-value = 0.012), ascites (*p*-value = 0.003), and higher levels of alpha-fetoproteins (*p*-value = 0.011). Carriers of the RAN*rs14035 (T) variant were more likely to develop HCC under allelic (OR = 1.76, *p*-value = 0.003) and recessive (OR = 3.27, *p*-value < 0.001) models. Our results suggest that *XPO5*rs34324334* and *RAN*rs14035* variants are independent risk factors for developing HCC.

## 1. Introduction

Hepatocellular carcinoma (HCC) is the second-highest cause of cancer-related mortality diagnosed worldwide [1]. It is the most challenging carcinoma to treat, and rates continue to grow rapidly: Global incidence is projected to reach over one million cases by 2025 [2,3]. The prevalence of HCC in Africa is highest among Egyptian individuals, which may be due to various genetic factors or to Egypt’s high hepatitis C virus (HCV) burden [4,5,6]. Although studies have sought to explain the involvement of hereditary and epigenetic modifications in the course of the disease, the molecular pathogenesis of HCC remains poorly understood [7,8].

MicroRNAs (miRNAs) are a subclass of small non-coding RNAs. Their ability to interact with mRNAs, which regulate cell proliferation, migration, and infiltration, has led to speculation that they are involved in the progression of HCC [9]. Genetic variations within the biogenesis of miRNA machinery may be associated with various carcinomas, including HCC [10]. miRNA biogenesis is a sophisticated process that begins intranuclearly with transcription of pri-miRNAs that submitted to modifications by microprocessors, forming pre-miRNAs. The export of miRNAs to the cytoplasm, with the aid of exportin 5 (XPO5) and its partner RanGTP, is believed to be correlated with limiting the rate of translocation of these miRNAs [11,12]. In the cytoplasm, miRNAs are shuttled via ribonuclease III (Dicer) and the transactivation response element RNA-binding protein (TRBP), which are incorporated into RNA-induced silencing complex (RISC) with Argonaute proteins [13,14,15].

The exportin 5 (XPO5) gene is situated at chromosome 6p21.1 with 32 exons, while RanGTP (RAN) is located at chromosome 12q24.33 with 7 exons and is thought to be one of the nucleocytoplasmic transporters belonging to the karyopherin β family [16,17]. The XPO5 gene encoded a specific protein known as exportin 5 protein that is responsible for the export of pri-miRNAs to the cytoplasm through the nuclear membrane that elucidates the critical function of this protein in the shuttling mechanism of different miRNAs. Thus, the genetic variations of the XPO5 gene could imply various epigenetic alterations and posttranscriptional modifications that altered the expression patterns of the miRNAs, leading to the progression of different cancer diseases [18]. Additionally, the transport of pre-miRNAs to cytosol triggers hydrolysis of RanGTP to RanGDP, mediated by RANBP1 and RANGAP1, respectively [19,20,21].

RAN (ras-related nuclear protein) is a small GTP-binding protein belonging to the RAS superfamily that is critical for protein transport through the nuclear pore complex [22]. RanGTP provokes microtubule assembly and is involved in normal mitotic spindle assembly and chromosome segregation [23]. It can also inhibit human vaccinia-related kinase (VRK1/2), which is involved in transcription regulation, nuclear membrane assembly, chromatin condensation, and the cell cycle [24]. RAN participates in DNA synthesis and, together with exportin 5, protects pre-miRNA from decay by nucleases, as it is surrounded by machinery on all sides [25,26]. Once pre-miRNA is translocated to the cytosol, RanGTP converts to RanGDP, causing conformational changes and releasing pre-miRNA [27].

During carcinogenesis, the dysregulation of the nuclear export of target pre-miRNAs could cause extensive expression of numerous abnormal miRNAs within tumor tissues, including miR-138 [28,29]. Additionally, the mutations in the XPO5 gene might disrupt the transportation process and inhibit miRNA maturation, causing more pre-miRNAs to be retained and trapped in the nucleus [30]. The mutated form of the XPO5 protein lacks the C-terminus region that has a crucial function in the joining of pre-miRNAs with XPO5 and RanGTP, causing the aggregation of the pre-miRNAs within the nucleus [31]. Single nucleotide variants (SNVs) within miRNA machinery genes, especially XPO5 and RAN, are widely reported to be associated with elevated risk of HCC, but the mechanism remains unclear [12,32]. Egypt has the highest incidence of HCC in the Middle East with nearly ~45.9/100,000 within males and ~22.7/100,000 within females. Additionally, HCC among Egyptians is considered the second-most prevalent carcinoma, after breast cancer [33,34]. The present work was designed to clarify the relationship of *XPO5* (*c.722G>A*; *rs34324334*) and *RAN* (*c.*770C>T*; *rs14035*) variants with susceptibility to HCC among Egyptian subjects from the Delta Region of Egypt. Additionally, we performed various bioinformatic analyses regarding the XPO5 and RAN genes to focus light on the importance of these potential variants in the progression of carcinomas.

## 2. Materials and Methods

### 2.1. Study Population

Our case–control design study comprised 234 participants [107 HCC patients and 127 healthy controls]; of these, HCC patients (87 [81.3%] male and 20 [18.7%] female) had a median age (interquartile range [IQR]) of 53.0 (45.0–61.0) years. HCC patients were diagnosed using multiphase magnetic resonance imaging (MRI) and/or computerized tomography (CT), in accordance with the guidelines of the American Association for the Study of Liver Disease (AASLD) [35]. The Institutional Review Board of the Faculty of Medicine at Tanta University in Tanta, Egypt, authorized the project (approval number 34230/11/20). All procedures performed in the study were in accordance with Helsinki Declaration guidelines. Recruitment and management of HCC patients were performed at the Outpatient Clinic of the Oncology Department at Tanta University Hospital from February 2021 to January 2022. A group of 127 cancer-free controls, matched for age (median 54.0 years, interquartile range [IQR] 49.0–58.0) and sex (94 [74.0%] male and 33 [26.0%] female), was recruited as volunteer blood donors. HCC patients with a history of other carcinomas, autoimmune disorders, diabetes, or kidney disease were excluded from the study. Clinical and demographic characteristics, including age, sex, age of onset, smoking status, splenomegaly status, and ascites status, were acquired from electronic medical health archives. 

Biochemical measurements, including hepatic aminotransferases, albumin, total bilirubin, INR (International Normalized Ratio), and creatinine, were determined using a fully automated chemical analyzer. Estimations of serological measurements, including hepatitis C virus autoantibodies (HCV Abs) and alpha-fetoprotein (AFP), were carried out using chemiluminescent techniques (Abbott PRISM Immunoassay Analyzer, Corston, Bath, UK). Hematological parameters were estimated with the aid of a multiparameter hematology counter system (CELL-DYN 3700 SL [Abbott Diagnostics]).

### 2.2. Genomic DNA Extraction and Amplification Analysis

Genomic DNA was extracted and purified with GeneJET Whole Blood Genomic DNA Purification Mini Kit (Thermo Fisher Scientific; Waltham, MA, USA). The concentration and absorbance of purified DNA was measured using a full spectrum NanoDrop^TM^ ND-1000 Spectrophotometer (NanoDrop Technologies, Wilmington, DE, USA). 

The allelic discrimination for the *XPO5* (*rs34324334*; *c.722G>A*) and *RAN* (*rs14035*; *c.*770C>T*) variants were determined using the polymerase chain reaction–restriction fragment length polymorphism (PCR-RFLP) method [36,37]. The electronic NCBI primer BLAST tool was applied to design the primers used in this work. The forward and reverse primers for the *XPO5* (*rs34324334*; *c.722G>A*) variant [F1: 5′-ATG AGA AGA CAC TCA GCG GCT C-3′, and R1: 5′-ACC TGT TAC CGA GGA CTT CAT G-3′] were processed to generate a PCR product of 218 bp. The two primers used for amplification of the *RAN* (*rs14035*; *c.*770C>T*) variant (F2: 5′-AAG CAG TGT TTG CTC CAC CTT C-3′, and R2: 5′-AGA ATT CCC AAC CTC CTG CC-3′) were designed to produce a PCR product of 216 bp. For these two variants, the PCR reaction was processed within a thermal cycler comprising 25 µL DreamTaq Green PCR Master Mix (2×) (Thermo Fisher Scientific), 1 µM forward primer, 1 µM reverse primer, 50 µL nuclease-free water, and 1 µg purified template DNA. The amplification reaction was adjusted with an initial denaturation at 94 °C for 5 min, followed by 35 cycles involving denaturation at 94 °C for 30 s, annealing at 60 °C for 60 s, and extension at 72 °C for 60 s, with a final extension at 72 °C for 7 min. The amplified products of the *XPO5* (*rs34324334*; *c.722G>A*) variant were subjected to endonuclease restriction enzyme Tsp45I (NmuCI) (New England Biolabs, Ipswich, MA, USA) and incubated at 65 °C for an hour, while the *RAN* (*rs14035*; *c.*770C>T*) variant was incubated with endonuclease restriction enzyme BseLI (Bs1I) (New England Biolabs) at 55 °C for 15 min. The digested fragments were electrophoresed using 2.5% agarose gel with ethidium bromide to allow for visualization.

### 2.3. Bioinformatic Analysis

Various online databases were explored for the identification of the functional roles of XPO5 and RAN along with their potential proteins, including the Ensembl database for chromosomal screening (https://www.ensembl.org/), National Center for Biotechnology Information (https://www.ncbi.nlm.nih.gov/gene/), compartments for localization of specific proteins inside the cell (https://www.proteinatlas.org/), Kaplan–Meier plotter database for survival analysis (https://kmplot.com/analysis/), protein data bank for the crystal structure of the specific proteins (https://www.rcsb.org/), string database for protein–protein interactions and gene ontology (https://string-db.org/), and GeneMANIA database for gene prediction (https://genemania.org/). All these electronic databases were accessed on 28 February 2023.

### 2.4. Statistical Analysis

Statistical analysis was conducted using STATA software version 17.0 (Stata Corporation, College Station, TX, USA). The detection of the degree of normal distribution was assessed by applying the Shapiro–Wilk test. Continuous data were assessed using the Wilcoxon rank-sum test, while categorical findings were evaluated using Fisher’s exact test and odds ratio (OR) with a 95% confidence interval [38,39]. The Hardy–Weinberg equilibrium for the *XPO5* (*rs34324334*; *c.722G>A*) and *RAN* (*rs14035*; *c.*770C>T*) variants was executed to identify the level of equilibrium among observed and expected values of HCC patients and cancer-free subjects [40]. Genetic association models were carried out using the electronic online tool for SNP analysis (www.snpstats.net, accessed on 25 October 2022). Multivariate analysis was performed and processed using R software version 4.2.2 with R studio version 2023.03 Build 386. Principal component analysis (PCA) was carried out using the “FactoMineR” and “Factoextra” packages. The statistical level was adjusted as a *p*-value of not more than 0.05.

## 3. Results

### 3.1. The Fundamental Characteristics of the Study Population

Generally, HCC patients showed a significantly higher correlation with tobacco smoking and weight (*p*-value < 0.05) compared to cancer-free controls. Of the HCC patients, 79 (73.8%) were cirrhotic, while 72 (67.3%) were ascitic. Biochemical, serological, and hematological measurements differed significantly between HCC patients and controls (*p*-value < 0.05) (Table 1).

### 3.2. XPO5*rs34324334 and RAN*rs14035 Variants with Susceptibility to HCC

The *XPO5*rs34324334* and *RAN*rs14035* variants approached Hardy–Weinberg equilibrium among cancer-free controls (*p*-value > 0.05). For the *XPO5*rs34324334* variant, the most prevalent genotype (A/A genotype) in HCC patients was 43%, compared to 3.2% in cancer-free controls (Figure 1A). The *XPO5*rs34324334* and *RAN*rs14035* variants among healthy subjects from the 1000 genome project dataset (https://www.internationalgenome.org/data, accessed on 28 February 2023) are summarized in Figure 1B,D. Principal component analysis (PCA) diagrams for HCC patients with *XPO5*rs34324334* and *RAN*rs14035* variants represented no distinct demarcation among different genotypes (Figure 1E,F). PCA categorized HCC patients into three clusters based on the genotype variants. Axis one and two for these tests accounted for 6.7% and 10.2% for the XPO5 genotypes and 7.1% and 10.2% for the RAN genotypes.

The minor allele frequency (A allele) was 57.5% among HCC patients compared to 11.8% among cancer-free controls (OR = 10.09, 95% CI = 6.32–16.11, *p*-value < 0.001; Table 2). The most prevalent genotype (T/T genotype) for the *RAN*rs14035* variant in HCC patients was 45.8%, compared to 21.3% in cancer-free controls (Figure 1C). The minor allele frequency was 56.5% in HCC patients and 42.5% in controls (OR = 1.76, 95% CI = 1.22–2.54, *p*-value < 0.001) (Table 2).

*XPO5*rs34334334* was significantly associated with HCC status among different genetic models, including heterozygote (OR = 4.73, 95% CI = 2.38–9.39, *p*-value < 0.001), homozygote (OR = 39.9, 95% CI = 13.2–121.1, *p*-value < 0.001), dominant (OR = 10.1, 95% CI = 5.46–18.4, *p*-value < 0.001), and recessive (OR = 24.1, 95% CI = 8.23–70.8, *p*-value < 0.001) (Table 3). The *RAN*rs14035* variant showed a significant association with elevated risk of HCC compared to cancer-free controls among different hereditary models, including homozygote (OR = 2.44, 95% CI = 1.27–4.68, *p*-value < 0.001) and recessive (OR = 3.27, 95% CI = 1.83–5.83, *p*-value < 0.001) (Table 3).

### 3.3. XPO5*rs34324334 and RAN*rs14035 Variants Stratified by Clinical and Laboratory Measurements among HCC Patients

The *XPO5*rs34324334* variant (A/A genotype) presented a significant association with hepatitis C cirrhosis (*p*-value = 0.012), ascites (*p*-value = 0.003), and higher levels of alpha-fetoproteins (*p*-value = 0.011) (Appendix A), while the *RAN*rs14035* variant (T/T genotype) failed to achieve statistical significance with any clinical or laboratory parameters (*p*-value > 0.05) (Figure 2).

### 3.4. In Silico Data Analysis

Computational bioinformatic analyses of the exportin 5 (XPO5) and ras-related nuclear protein (RAN) genes are summarized in Figure 3. The XPO5 gene has specific properties that indicate its coding protein comprising 1204 amino acid residues. This encoded protein is a major component of the nuclear export receptor complex along with RAN. The RAN gene encodes a specific protein with 216 amino acid residues. The *XPO5*rs34324334* (*c.722G>A*; *p.Ser241Asn*) variant is located on the seventh exon, while the *RAN*rs14035* (*c.*770C>T*) variant is situated on the last exon. Protein–protein networks indicated that XPO5 and RanGTP binding proteins were involved in various miRNA biological functions, including pre-miRNA export from the nucleus, miRNA metabolic processes, RNA transport, and the miRNA biogenesis pathway. Data provided from the GeneMANIA database for gene–gene interactions revealed the role of the XPO5-RanGTP complex in post-transcriptional gene silencing by RNA, miRNA gene silencing, nuclear export, and RNA localization from the nucleus. The protein atlas database identified the presence of these proteins in higher abundance in the nuclear membrane. The Kaplan–Meier plotter database detected low and high mRNA expression levels of XPO5 and RAN among patients with hepatic carcinoma, respectively.

## 4. Discussion

MicroRNAs (miRNAs) are a crucial category of noncoding RNAs, with about 22 nucleotides that are involved in the regulation mechanisms of hereditary and epigenetic pathways [41]. Each miRNA can bind to several mRNA targets, causing instability, degradation, and inhibition [42]. Generally, miRNAs regulate gene expression by attaching to the 3′ untranslated region (3′ UTR) of target mRNAs, but any alteration in expression of synthesized miRNA could affect hundreds of genes [43]. miRNA biogenesis genes and their variants are involved in the pathogenesis of several diseases, including HCC [12,44,45]. To the best of our knowledge, this work is the first to study the association of the *XPO5*rs34324334* and *RAN*rs14035* variants with increased risk of HCC among Egyptian subjects. The XPO5 gene belongs to the karyopherin β family, which plays a crucial role in shuttling from nucleus to cytoplasm in a RANGTP-dependent manner. The *XPO5*rs34324334* variant could represent a missense variant (*XPO5*p.Ser241Asn*) modulating the protein structure, which could reflect a change in its behavior associated with pre-miRNA nuclear export and a subsequent regulatory role in HCC.

In the present study, the *XPO5*rs34324334* (*c.722G>A*) variant showed a significant association according to the allelic model (OR = 10.09 and *p*-value < 0.001). Testing the inheritance model with frequency of genotypes revealed a significant effect on the risk of HCC in dominant and recessive models, with odds ratios of 10.1 and 24.1, respectively. We also found significant associations between the *XPO5*rs34324334* variant and cirrhotic liver and ascites status among HCC patients. These findings were consistent with a study that revealed a strong association between the *XPO5*rs34324334* variant and elevated risk of breast carcinoma in postmenopausal women with breast cancer (dominant model: OR = 1.76, *p*-value < 0.05) [46]. The *XPO5*rs34324334* variant is thought to be in a highly conserved exportin-1/importin-b-like region and may affect the development of carcinomas. To test and predict the changes in protein stability upon mutation, we used DynaMut (https://biosig.lab.uq.edu.au/dynamut/; accessed on 20 December 2022) to analyze the modification in the *XPO5* (*rs34324334*; *p.Ser241Asn*) missense variant and identified a destabilization effect (ΔΔG = −0.236 kcal/mol) on the functional mechanism of the XPO5 protein. Lack of XPO5 expression and mutated forms of XPO5 are both highly correlated with reduced levels of miRNAs and their inhibitory targets; restoring functional XPO5 seemed to act as a tumor suppressor. XPO5 mutations subsequently disrupt its role as a tumor suppressor gene and modulator for cancer through attenuated production of miRNAs that, in turn, promote growth regulatory gene TGGBR2 as well as proapoptotic gene BAX. Moreover, the aberration of XPO5 is linked to a global reduction in miRNAs and subsequent altered regulation [30,47].

The RAN gene acts as a powerhouse for the pre-miRNA nuclear shuttling process via exportin 5. Our *RAN*rs14035* variant data suggest an association within the allelic model (T allele vs. C allele; odds ratio 1.76 and *p*-value < 0.05). Upon testing the inheritance model, this 3′ UTR variant was associated with a significantly elevated risk of HCC under homozygote and recessive comparisons (OR = 2.44 and 3.27, respectively). Similar results were found in a study among Asian patients diagnosed with HCC, which revealed a strong correlation with elevated risk of hepatic cancer under a recessive model (OR = 2.54) [32]. Another report, among Korean patients with colorectal cancer (CRC), indicated that the *RAN*rs14035* variant was significantly correlated with a decreased risk of CRC (OR = 0.690; *p*-value = 0.016) [48]. Several meta-analysis reports addressed the contribution of the RAN*rs14035 variant to cancer development among various ethnic populations [49,50,51]. Another study revealed a significant association between the *RAN*rs14035* variant and an increased risk of end-stage renal disease among Egyptian subjects (OR = 5.18) [44], whereas the *RAN*rs14035* variant showed no association with elevated risk of various types of carcinomas, including hepatocellular carcinoma [45] and renal cell carcinoma [52].

Testing the functional role of the regulatory non-coding region within the *RAN*rs14035* variant using the RegulomeDB electronic tool (https://regulomedb.org/, accessed on 20 December 2022) resulted in a score of 0.59 with a moderate shift toward 1, suggesting that alterations within the *RAN*rs14035* variant may modify the stability of various mRNAs, particularly miR-575 and miR-182, as has been previously suggested for larynx cancer [53]. Our results were consistent with a comprehensive meta-analysis that confirmed the contribution of RAN*rs14035 to cancer risk [54]. Additionally, knocking down XPO5 expression resulted in a reduction in global miRNA levels, thus enhancing carcinogenesis [55].

Another tool for exploring annotations of noncoding variants, HaploReg, revealed overlap between *RAN*rs14035* and nearby variants as well as interactions with PLZF, ZEB1, and ZBTB12 motifs (Appendix A). The promyelocytic leukemia zinc finger protein (PLZF) contributes to tumor formation by regulating cell growth, differentiation, and apoptosis, in addition to its role regulating cytokine production to ameliorate cancer progression [56,57,58]. Zinc finger E-box-binding homeobox 1 (ZEB1) is involved in epithelial-to-mesenchymal transition (EMT) and is significantly associated with poorer cancer prognosis, including for HCC [59]. *RAN*rs14035* may contribute to changes in the expression of RAN itself, binding of miRNAs, and interaction with network motifs, all of which can lead to oncogenesis.

## 5. Conclusions

Evidence suggests that the genes XPO5 and RAN may contribute to the development of tumors, including HCC, and may be involved in the mechanism by which microRNAs (miRNAs) are synthesized in cancer. This is the first work studying the association of *XPO5*rs34324334* and *RAN*rs14035* variants with elevated risk of HCC among Egyptian subjects. We found evidence suggesting that these genetic variants are associated with a variety of indicators of poor liver health, including HCC. Our data strongly suggests that *XPO5*rs34324334* and *RAN*rs14035* variants could represent independent risk factors for developing HCC. Genetic testing for these variants in HCC patients may be a useful tool for predicting which patients are at a higher risk of aggressive characteristics and poorer outcomes, allowing clinicians to choose more personalized and effective treatments early in the course of the disease when treatment is most likely to be effective.

## Figures and Tables

**Figure 1 jpm-13-00959-f001:**
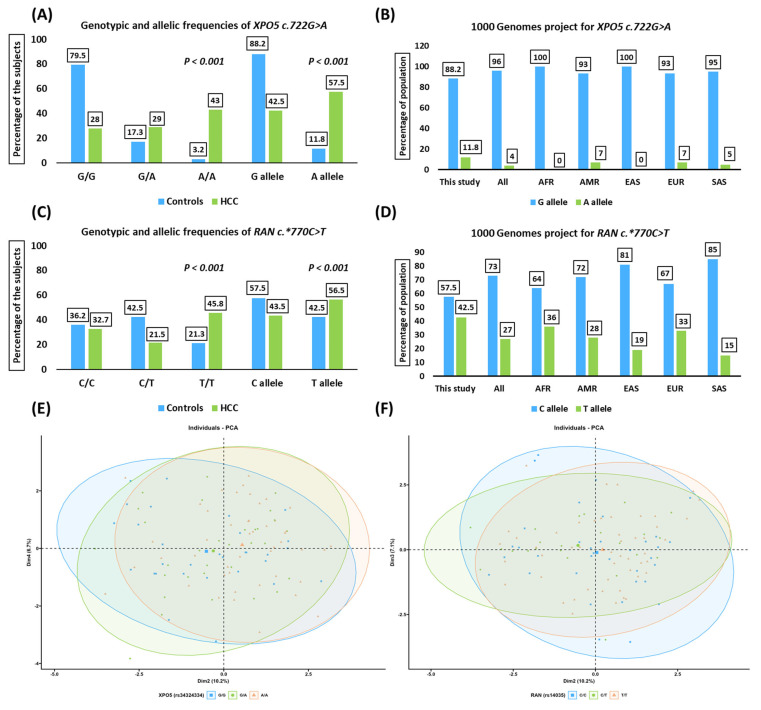
Genotypic and allelic frequencies of the study population. (**A**) Genotypic and allelic distribution of the *XPO5* (*rs34324334*; *c.722G>A*) variant among HCC patients compared to cancer-free controls. (**B**) The 1000 genome project for the *XPO5* (*rs34324334*; *c.722G>A*) variant. (**C**) Genotypic and allelic distribution of the *RAN* (*rs14035*; *c.*770C>T*) variant among HCC patients compared to cancer-free controls. (**D**) The 1000 genome project for the *RAN* (*rs14035*; *c.*770C>T*) variant. (**E**,**F**) Principal component analysis (PCA) of the *XPO5* (*rs34324334*; *c.722G>A*) and *RAN* (*rs14035*; *c.*770C>T*) variants among HCC patients. These figures present no distinct demarcation of the HCC patients with different genotypes. AFR, Africa; AMR, America; EAS, East Asia; EUR, Europe; SAS, South Asia; HCC, hepatocellular carcinoma.

**Figure 2 jpm-13-00959-f002:**
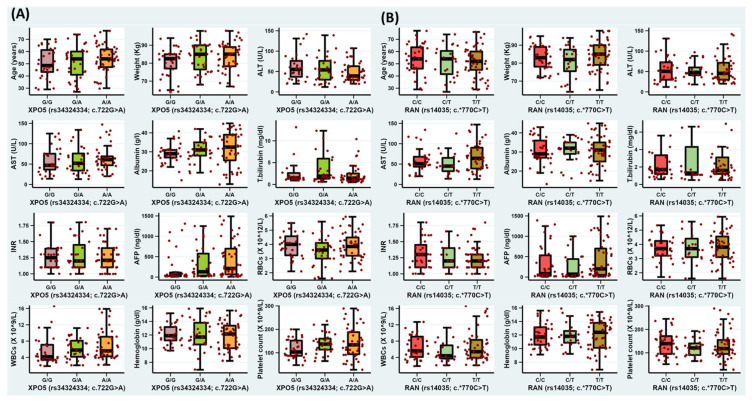
The impact of *XPO5*rs34324334* and *RAN*14035* variants on serological, biochemical, and hematological variables. (**A**) Box plot for the *XPO5*rs34324334* variant among HCC patients; (**B**) box plot for the *RAN*14035* variant among HCC patients.

**Figure 3 jpm-13-00959-f003:**
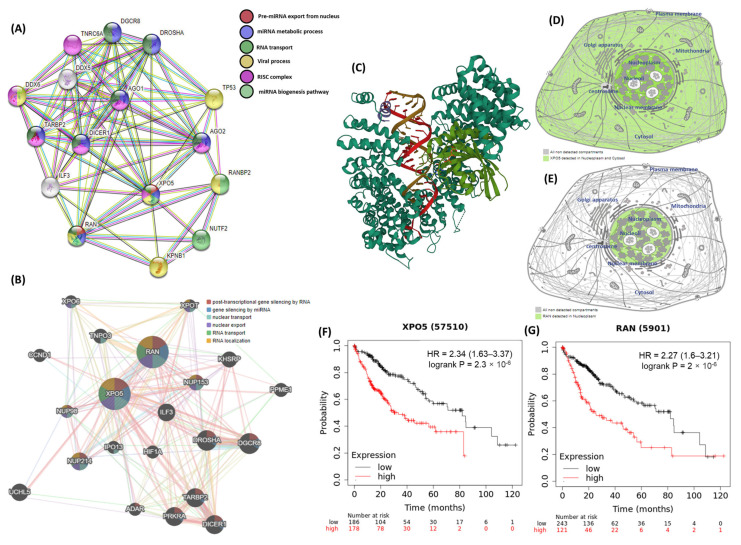
The bioinformatic framework analysis of the XPO5 and RAN genes. (**A**) protein–protein interactions of the XPO5 and RanGTP proteins using the STRING database. (**B**) gene–gene interactions using the GeneMANIA database. (**C**) the crystal structure of the XPO5:RanGTP:pre-miRNA complex. (**D**) subcellular localization of the XPO5 protein, with darker colors indicating more abundance. (**E**) subcellular localization of the RanGTP protein. (**F**,**G**) survival analysis data for high and low XPO5 and RAN expression. (Data source: The Human Protein Atlas, GeneMANIA, STRING version 11.0, and Kaplan–Meier plotter database).

**Table 1 jpm-13-00959-t001:** The main demographic, clinical, and laboratory variables of the study participants.

Variable	Levels	Cancer-Free Controls	HCC Patients	*p*-Value
		**(n = 127)**	**(n = 107)**	
**Demographic and Clinical Characteristics**				
Age, years	Median (IQR)	54.0 (49.0–58.0)	53.0 (45.0–61.0)	0.832
	≤40 years	15 (11.8)	21 (19.6)	0.106
	>40 years	112 (88.2)	86 (80.4)	
Weight, kg	Median (IQR)	80.0 (76.0–87.0)	84.0 (78.0–89.0)	**0.014**
Gender	Males	94 (74.0)	87 (81.3)	0.211
	Females	33 (26.0)	20 (18.7)	
Smoking	Positive	17 (13.4)	32 (29.9)	**0.002**
	Negative	110 (86.6)	75 (70.1)	
Consanguinity	Positive	---	26 (24.3)	NA
	Negative	---	81 (75.7)	
Cirrhotic liver	Positive	---	79 (73.8)	NA
	Negative	---	28 (26.2)	
Hypertension	Positive	---	33 (30.8)	NA
	Negative	---	74 (69.2)	
Ascites status	Presence	---	72 (67.3)	NA
	Absence	---	35 (32.7)	
Ascites grade	Grade 1 (Mild)	---	7 (6.5)	NA
	Grade 2 (Moderate)	---	18 (16.8)	
	Grade 3 (Large)	---	47 (43.9)	
Splenomegaly	Presence	---	93 (86.9)	NA
	Absence	---	14 (13.1)	
Splenomegaly classes	Mild enlarged	---	40 (37.4)	NA
	Moderate enlarged	---	44 (41.1)	
	Massive enlarged	---	9 (8.4)	
**Biochemical Measurements**				
ALT, U/L	Median (IQR)	29.0 (21.0–35.0)	52.0 (33.0–85.0)	**<0.001**
AST, U/L	Median (IQR)	26.0 (21.0–31.0)	64.0 (42.0–115.0)	**<0.001**
Albumin, g/L	Median (IQR)	42.0 (38.0–47.0)	31.0 (27.0–36.0)	**<0.001**
Total bilirubin, mg/dL	Median (IQR)	0.90 (0.70–1.00)	1.60 (1.10–3.50)	**<0.001**
Direct bilirubin, mg/dL	Median (IQR)	0.21 (0.15–0.26)	0.84 (0.47–2.12)	**<0.001**
Indirect bilirubin, mg/dL	Median (IQR)	0.67 (0.51–0.84)	0.81 (0.56–1.30)	**<0.001**
INR	Median (IQR)	1.00 (1.00–1.10)	1.20 (1.10–1.40)	**<0.001**
Creatinine, mg/dL	Median (IQR)	0.90 (0.70–1.00)	1.00 (0.80–1.30)	**<0.001**
**Serological Investigations and Tumor Markers**				
Anti-HCV	Positive	0.0 (0.0	87 (81.3)	**<0.001**
	Negative	127 (100.0)	20 (18.7)	
AFP, ng/mL	Median (IQR)	7.0 (4.0–9.4)	115.0 (32.0–582.0)	**<0.001**
**Hematological Parameters**				
WBCs, ×10^9^/L	Median (IQR)	7.8 (5.8–9.4)	5.3 (3.9–8.5)	**<0.001**
RBCs, ×10^12^/L	Median (IQR)	4.5 (4.1–5.0)	3.7 (3.1–4.3)	**<0.001**
Hematocrit (HCT), %	Median (IQR)	37.5 (34.1–42.0)	34.9 (30.2–39.9)	**<0.001**
Hemoglobin, g/dL	Median (IQR)	13.5 (12.8–15.0)	12.0 (10.5–13.6)	**<0.001**
Platelet count, ×10^9^/L	Median (IQR)	258.0 (198.0–341.0)	127.0 (92.0–164.0)	**<0.001**

Data are presented as numbers with percentages or median with interquartile range (IQR). Fisher’s exact and two-sample Wilcoxon rank-sum tests were applied. HCC: hepatocellular carcinoma; ALT: alanine transaminase; AST: aspartate transaminase; INR: international normalized ratio; Anti-HCV: hepatitis C virus antibodies; AFP: alpha-fetoprotein; WBCs: white blood cells; RBCs: red blood cells. **Bold** indicates *p*-value < 0.05.

**Table 2 jpm-13-00959-t002:** Genotypic and allelic frequencies of the *XPO5*rs34324334* and *RAN*rs14035* variants among the study population.

Genetic Polymorphisms	Cancer-Free Controls	HCC Patients	OR (95% CI)	*p*-Value
***XPO5* (*rs34324334*; *c.722G>A*)**				
Genotypic frequencies	n (%) 127	n (%) 107		
G/G	101 (79.5)	30 (28.0)	1.0	
G/A	22 (17.3)	31 (29.0)	**4.75 (2.40–9.38)**	**<0.001**
A/A	4 (3.2)	46 (43.0)	**38.7 (12.9–116.3)**	**<0.001**
HWE	χ^2^ = 3.40, *p* = 0.058	χ^2^ = 17.75, *p* < 0.001		
Allelic frequencies	n (%) 254	n (%) 214		
G allele	224 (88.2)	91 (42.5)	1.0	**<0.001**
A allele	30 (11.8)	123 (57.5)	**10.09 (6.32–16.11)**	
***RAN* (*rs14035*; *c.*770C>T*)**				
Genotypic frequencies	n (%) 127	n (%) 107		
C/C	46 (36.2)	35 (32.7)	1.0	
C/T	54 (42.5)	23 (21.5)	0.56 (0.29–1.08)	0.099
T/T	27 (21.3)	49 (45.8)	**2.38 (1.25–4.54)**	**0.010**
HWE	χ^2^ = 2.15, *p* = 0.142	χ^2^ = 33.9, *p* < 0.001		
Allelic frequencies	n (%) 254	n (%) 214		
C allele	146 (57.5)	93 (43.5)	1.0	**0.003**
T allele	108 (42.5)	121 (56.5)	**1.76 (1.22–2.54)**	

Data are presented as numbers with percentages. Fisher’s exact test was applied. Abbreviations: OR, odds ratio; CI, confidence interval; HWE, Hardy–Weinberg equilibrium. **Bold** values indicate *p*-value < 0.05.

**Table 3 jpm-13-00959-t003:** Genetic association models of the *XPO5*rs34324334* and *RAN*rs14035* variants with the risk of hepatocellular carcinoma.

Model	Genotypes	Cancer-Free Controls	HCC Patients	Crude OR (95% CI)	*p*-Value	Adjusted OR (95% CI)	*p*-Value
** *XPO5*rs34324334* **		**n (%) 127**	**n (%) 107**				
Codominant	G/G	101 (79.5)	30 (28.0)	1.0	**<0.001**	1.0	**<0.001**
	G/A	22 (17.3)	31 (29.0)	**4.74 (2.40–9.38)**		**4.73 (2.38–9.39)**	
	A/A	4 (3.2)	46 (43.0)	**38.7 (12.9–116.3)**		**39.9 (13.2–121.1)**	
Dominant	G/G	101 (79.5)	30 (28.0)	1.0	**<0.001**	1.0	**<0.001**
	G/A + A/A	26 (20.5)	77 (72.0)	**9.97 (5.45–18.22)**		**10.1 (5.46–18.4)**	
Recessive	G/G + G/A	123 (96.8)	61 (57.0)	1.0	**<0.001**	1.0	**<0.001**
	A/A	4 (3.2)	46 (43.0)	**23.2 (7.98–67.4)**		**24.1 (8.23–70.8)**	
Log-additive	---	---	---	**5.68 (3.61–8.94)**	**<0.001**	**5.73 (3.63–9.05)**	**<0.001**
** *RAN/rs14035* **		**n (%) 127**	**n (%) 107**				
Codominant	C/C	46 (36.2)	35 (32.7)	1.0	**<0.001**	1.0	**<0.001**
	C/T	54 (42.5)	23 (21.5)	0.56 (0.29–1.08)		0.53 (0.27–1.03)	
	T/T	27 (21.3)	49 (45.8)	**2.39 (1.25–4.54)**		**2.44 (1.27–4.68)**	
Dominant	C/C	46 (36.2)	35 (32.7)	1.0	0.570	1.0	0.610
	C/T + T/T	81 (63.8)	72 (67.3)	1.17 (0.68–2.01)		1.15 (0.67–1.99)	
Recessive	C/C + C/T	100 (78.7)	58 (54.2)	1.0	**<0.001**	1.0	**<0.001**
	T/T	27 (21.3)	49 (45.8)	**3.13 (1.77–5.53)**		**3.27 (1.83–5.83)**	
Log-additive	---	---	---	**1.53 (1.11–2.11)**	**0.009**	**1.54 (1.11–2.13)**	**0.008**

Data are presented as numbers with percentages. Chi squared test was applied. Adjusted by age and gender. Abbreviations: OR, Odds Ratio; CI, Confidence Interval. **Bold** values indicate *p*-value < 0.05.

## Data Availability

Data are available from the corresponding author upon reasonable request. Sources of in silico data analysis sections are available in public repositories through the links provided in the manuscript.

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
