# Peer review of "Unleash Multifunctional Role of miRNA Biogenesis Gene Variants (XPO5*rs34324334 and RAN*rs14035) with Susceptibility to Hepatocellular Carcinoma"

_jpm, 2023, doi:10.3390/jpm13060959_

Round 1

Reviewer 1 Report

There is a lack of connection between the introduction paragraphs.

There should be a short description of XPO5 to make its link in the given context, in the introduction.

Why this study is especially being done in the Egyptian cohort, could be explained better in aims.

107 patients, and what about the other 127 participants? If they were control samples, it needs to be added in the initial sentences of the study population.

Also, it should be mentioned in the study population that, were all participants Egyptian only.

Reference is missing in the last sentences of the first paragraph of the discussion.

The conclusion could be elaborated more by giving some conclusive remarks on XPO5 and RAN variants in terms of risk, or maybe the conclusion could be merged with the discussion if couldn't be elaborated.

The rest of the things have been addressed in the Recommendations for the authors.

There are some minor mistakes throughout the manuscript which could be improved after a thorough reading of the paper.

Author Response

Reviewer #1

Comment #1

There is a lack of connection between the introduction paragraphs. There should be a short description of XPO5 to make its link in the given context, in the introduction.

Response #1

Thanks for the time and effort of the respected reviewer. We updated the introduction based on the reviewer suggestion with focus on the connection between paragraphs and illustrate a more description of the XPO5 gene with the progression of cancer.

We have demonstrated that our manuscript is laid across the five paragraphs. Firstly, we highlighted the crucial dilemma of HCC in the first paragraph. Then, we pointed out the regulatory roles of miRNAs in regulation and cancer in addition to their biogenesis machinery in the second one which also gained much emphasis in the third and fourth paragraph especially XPO5 and RanGTP genes, respectively.

Lastly, in the fifth paragraph showed that carcinogenesis is tightly correlated with biogenesis machinery and its alterations with our aims to clarify the relationship of XPO5 (c.722G>A; rs34324334) and RAN (c.*770C>T; rs14035) variants with susceptibility to HCC. In conclusion, you can observe that five paragraphs represent the pipeline of our work starting from disease issue, underlying machineries, and alterations that might exhibit an impact leading to HCC risk. In addition, we added this part in the introduction section regarding the XPO5.

 The XPO5 gene encoded a specific protein known as exportin 5 protein that is responsible for pri-miRNAs export to cytoplasm through the nuclear membrane that elucidates the critical function of this protein in the shuttling mechanism of different miRNAs. Thus, the genetic variations of the XPO5 gene could imply various epigenetic alterations, and posttranscriptional modifications that altered the expression patterns of the miRNAs leading to the progression of different cancer diseases (16).

 Comment #2

Why this study is especially being done in the Egyptian cohort, could be explained better in aims. 107 patients, and what about the other 127 participants? If they were control samples, it needs to be added in the initial sentences of the study population. Also, it should be mentioned in the study population that, were all participants Egyptian only.

Response #2

Thanks for the reviewer time. In fact, this design work was constructed to assess the contribution of the XPO5 (c.722G>A; rs34324334) and RAN (c.*770C>T; rs14035) variants with the increased risk of HCC among Egyptian subjects and based on the reviewer suggestion, we modified our aim to be more relevant. Additionally, this work was designated based on 234 participants (107 HCC patients and 127 healthy controls) from the Delta region of Egypt and all the participants were of Egyptian ancestry only. Moreover, regarding to why this study was performed among Egyptian cohort because of Egypt is characterized with greatest incidence of HCC in middle east with ~45.9/100 000 in males and ~22.7/100 000 in females, high incidence could be correlated with the high prevalence of cirrhosis-related-HCV which had been epidemic until the recent years therapy campaign. Additionally, among different carcinomas prevalence, HCC in Egypt is the 2nd most prevalent cancer in males after bladder cancer as well as the 2nd most prevalent in women after breast cancer. The present study was conducted in Egypt as part of MSc research project and ethical approval was obtained from Review Board of the Faculty of Medicine at Tanta University in Tanta, Egypt authorized the project (approval number 34230/11/20), given in materials and methods.

 Comment #3

Reference is missing in the last sentences of the first paragraph of the discussion.

Response #3

Thanks to the reviewer hint, we added the reference in the first paragraph in the discussion section.

 Comment #4

The conclusion could be elaborated more by giving some conclusive remarks on XPO5 and RAN variants in terms of risk, or maybe the conclusion could be merged with the discussion if couldn't be elaborated.

Response #4

Thanks for the reviewer suggestion, we expanded the conclusion with more focus on the roles of these potential genes in the course of the disease.

 Evidence suggests that the genes XPO5 and RAN may contribute to the development of tumors, including HCC, and may be involved in the mechanism by which microRNAs (miRNAs) are synthesized in cancer. This is the first work studying the association of XPO5*rs34324334 and RAN*rs14035 variants with elevated risk of HCC among Egyptian subjects. We found evidence suggesting that these genetic variants are associated with a variety of indicators of poor liver health, including HCC. Our data strongly suggest that XPO5*rs34324334 and RAN*rs14035 variants could represent independent risk factors for developing HCC.  Genetic testing for these variants in HCC patients may be a useful tool for predicting which patients are at higher risk for aggressive characteristics and poorer outcomes, allowing clinicians to choose more personalized and effective treatments early in the course of the disease when treatment is most likely to be effective.

 Comment #5

The rest of the things have been addressed in the Recommendations for the authors.

Response #5

Thanks for the reviewer effort and time.

 Comment #6

There are some minor mistakes throughout the manuscript which could be improved after a thorough reading of the paper.

Response #6

Thanks for the time and effort of the respected reviewer. We revised the whole manuscript with the assistance of Loula Burton from Tulane’s Research Proposal Development Office to avoid grammatical errors and improve the quality of this work. Additionally, we mentioned this part in the acknowledgments section.

Acknowledgments: A sincere thank you to Loula Burton from Tulane’s Research Proposal Development Office for her diligent editing and proofreading of this paper.

Reviewer 2 Report

Review -  Unleash multifunctional role of miRNA biogenesis gene vari-2 ants (XPO5*rs34324334 and RAN*rs14035) with susceptibility to 3 hepatocellular carcinoma

 Line 46 – That are submitted to

Line 55 – Missing reference after karyopherin β family

Line 99 – Please state what INR stands for

Line 101 – Could you state the name of the chemiluminescent techniques?

Line 110 – Instead of manipulated, change to determined

Line 119 - Do you mean SYBR green?

Line 143 compared to cancer-free controls. Then the next line, state that this is from the HCC patients as now you mention this is from the total patients, which is not the case. After looking at the table this is clear, however not in text, Maybe something along the lines “Of the HCC patients, 79 were cirrhotic patients and 72 has ascites.

 In table 1, I would suggest to remove the Chi-square column, here is why, you have some significance there, but it does not really tell which group against which group is significant. For instance is smoking,  

 In table 1 in the footnote, add (IQR) in this sentence: Data are presented as numbers with percentages or median with interquartile range. The you don’t have to add it afterwards.

 In figure 1. Add titles to the Y axis in panel A and X, is that percentage of samples/patients?

If I understand correctly the most common allele in HCC was A allele with 57.5 whereas in cancer free patients A allele represented 11.8. In the text the number are expressed differently 57.5 is written as 0.575 in HCC patients and 0.118 in cancer free controls, please choose one.

 I think you could reorder your results by first describing your figure 1 and then go to the results in table 2. At the moment results are intercalated.

 I would suggest to move this text “XPO5*rs34334334 was significantly associated with HCC status among different genetic models including heterozygote (OR=4.73, 95% CI=2.38–9.39, p-value < 0.001), homozygote (OR=39.9, 95% CI=13.2–121.1, p-value < 0.001), dominant (OR=10.1, 95% CI=5.46–18.4, p-value < 0.001), and recessive (OR=24.1, 95% CI=8.23–70.8, p-value < 0.001) (Table 3).

 The RAN*rs14035 variant showed a significant association with elevated risk of HCC compared to cancer-free controls among different hereditary models including homozygote (OR=2.44, 95% CI=1.27–4.68, p-value < 0.001), and recessive (OR=3.27, 95% CI=1.83-5.83, p-value < 0.001) (Table 3).” After table 2.

 Line 207 XPO5 and RAN definitions are already defined in line 53

Line 209 - IS located

Line 275 – State what CRC stands for

 In discussion, I think there is opportunity to expand the comparison of your results with others. For instance, expand on the comparison with references 27 and 41 which are in line 265,

 In line 288 you refer to reference 47 however this reference is about larynx cancer, you could add something like “ as it has been previously shown/suggested in larynx cancer”

 Here are some references that could be used to strengthen your discussion:

 https://www.ncbi.nlm.nih.gov/pmc/articles/PMC6985935/

 https://pubmed.ncbi.nlm.nih.gov/24676133/

 https://pubmed.ncbi.nlm.nih.gov/29226993/

 Moderate modification, I have indicated where I have identified some errors. 

Author Response

Unleash multifunctional role of miRNA biogenesis gene vari-2 ants (XPO5*rs34324334 and RAN*rs14035) with susceptibility to 3 hepatocellular carcinoma.

Comment #1

Line 46 – That are submitted to.

Response #1

Thanks for the reviewer time. We followed the reviewer suggestion.

 Comment #2

Line 55 – Missing reference after karyopherin β family

Response #2

We added the requested reference.

 Comment #3

Line 99 – Please state what INR stands for.

Response #3

Thanks for the reviewer time. INR standing for International normalized ratio.

 Comment #4

Line 101 – Could you state the name of the chemiluminescent techniques?

Response #4

Thanks for the reviewer hint. We added the name of the chemiluminescent instrument using immune assay.

 Comment #5

Line 110 – Instead of manipulated, change to determined.

Response #5

Thanks for the recommendations, we followed the reviewer advice.

 Comment #6

Line 119 - Do you mean SYBR green?

Response #6

Thanks for reviewer query, in fact, we used DreamTaq Green PCR Master Mix (2X) in our conventional PCR technique to be used in genotyping and amplification as described in the following link.

https://www.thermofisher.com/order/catalog/product/K1081

 Comment #7

Line 143 compared to cancer-free controls. Then the next line, state that this is from the HCC patients as now you mention this is from the total patients, which is not the case. After looking at the table this is clear, however not in text, maybe something along the lines “Of the HCC patients, 79 were cirrhotic patients and 72 has ascites.

Response #7

Thanks for the reviewer time, we followed all the recommendations.

 Comment #8

In table 1, I would suggest removing the Chi-square column, here is why, you have some significance there, but it does not really tell which group against which group is significant. For instance, is smoking.

Response #8

Thanks for the reviewer hint. In table (1), we demonstrated the demographic, clinical data, and laboratory findings between HCC patients and cancer-free controls and did not use chi square, but instead with the aid of Fisher’s exact test between the two groups.

 Comment #9

In table 1 in the footnote, add (IQR) in this sentence: Data are presented as numbers with percentages or median with interquartile range. The you don’t have to add it afterwards.

Response #9

We followed the reviewer suggestion.

 Comment #10

In figure 1. Add titles to the Y axis in panel A and X, is that percentage of samples/patients?

Response #10

We followed the reviewer’s suggestion and added the Y axis.

 Comment #11

If I understand correctly the most common allele in HCC was A allele with 57.5 whereas in cancer free patients A allele represented 11.8. In the text the number are expressed differently 57.5 is written as 0.575 in HCC patients and 0.118 in cancer free controls, please choose one.

Response #11

We followed the reviewer’s advice and modified the values.

  Comment #12

I think you could reorder your results by first describing your figure 1 and then go to the results in table 2. At the moment results are intercalated. I would suggest to move this text “XPO5*rs34334334 was significantly associated with HCC status among different genetic models including heterozygote (OR=4.73, 95% CI=2.38–9.39, p-value < 0.001), homozygote (OR=39.9, 95% CI=13.2–121.1, p-value < 0.001), dominant (OR=10.1, 95% CI=5.46–18.4, p-value < 0.001), and recessive (OR=24.1, 95% CI=8.23–70.8, p-value < 0.001) (Table 3). The RAN*rs14035 variant showed a significant association with elevated risk of HCC compared to cancer-free controls among different hereditary models including homozygote (OR=2.44, 95% CI=1.27–4.68, p-value < 0.001), and recessive (OR=3.27, 95% CI=1.83-5.83, p-value < 0.001) (Table 3).” After table 2.

Response #12

We followed the reviewer’s recommendations.

 Comment #13

Line 207 XPO5 and RAN definitions are already defined in line 53.

Response #13

We followed the reviewer’s advice.

 Comment #14

Line 209 - IS located

Response #14

We followed the reviewer’s modifications.

 Comment #15

Line 275 – State what CRC stands for

Response #15

Thanks to the reviewer time, CRC stands for colorectal cancer.

 Comment #16

In discussion, I think there is an opportunity to expand the comparison of your results with others. For instance, expand on the comparison with references 27 and 41 which are in line 265.

 Response #16

We expanded the comparison, as addressed in (Lund et al., 2004; Melo et al., 2010).

XPO5 mutations subsequently disrupt its role as tumor suppressor gene as modulator for cancer through attenuated production of miRNAs that in turn promote growth regulatory gene TGGBR2 as well as proapoptotic gene BAX. Also, the aberration of XPO5 is linked to global reduction of miRNAs and subsequent altered regulation.”

 Comment #17

In line 288 you refer to reference 47 however this reference is about larynx cancer, you could add something like “ as it has been previously shown/suggested in larynx cancer”

Response #17

We followed the reviewer’s modifications.

  Comment #18

Here are some references that could be used to strengthen your discussion:

 https://www.ncbi.nlm.nih.gov/pmc/articles/PMC6985935/

 https://pubmed.ncbi.nlm.nih.gov/24676133/

 https://pubmed.ncbi.nlm.nih.gov/29226993/

Response #18

 Thanks for the reviewer’s suggestions, we added this part in the discussion section:

“Our results were consistent with comprehensive meta-analysis that confirmed the contribution of the RAN*rs14035 with cancer risk. Additionally, knocking down XPO5 expression resulted in the reduction of global miRNA levels, thus enhancing carcinogenesis.”

 Comment #19

Moderate modification, I have indicated where I have identified some errors.

Response #19

Thanks for the time and effort of the respected reviewer. We revised the whole manuscript with the assistance of Loula Burton from Tulane’s Research Proposal Development Office to avoid grammatical errors and improve the quality of this work. Additionally, we mentioned this part in the acknowledgments section.

Reviewer 3 Report

This study aims to elucidate the association of specific XPO5 and RAN variants (c.722G>A; rs34324334 and c.*770C>T; rs14035, respectively) with susceptibility to HCC in an Egyptian cohort. The results are highly interesting and of potential significant impact. Nevertheless, several aspects of the methodology need additional detail to improve the clarity and replicability of the study.

1. The manner in which genotype and allele frequencies for the XPO5rs34324334 and RANrs14035 variants were derived from the 1000 Genomes Project is not elucidated. A clearer description of this process is necessary.

2. The methods utilized to perform the PCA) on HCC patients carrying the XPO5rs34324334 and RANrs14035 variants require further explanation. Details about the variables included in the PCA, any data preprocessing steps undertaken, and the software or software package employed for the PCA would be beneficial.

3. More thorough details concerning the steps for the bioinformatics analysis of the XPO5 and RAN genes are needed in the method part. This should include information on the identification of genes, the analysis of protein-protein and gene-gene interactions, the determination of subcellular protein localization, and the data collection methods for survival analysis.

Besides that, Line 79: while the overall goal of the study is mentioned, it would be beneficial to articulate the specific purpose or hypothesis of the study more directly.

L164-166, regarding the PCA (Figure 1E & 1F) with "no distinct demarcation among different genotypes" might need further explanation or clarification. Does this mean that these genetic variants do not stratify the patients into different groups?

The manuscript is generally understandable, but there are several points that could improve its quality.

For example, a sentence in the introduction linking the paragraph on miRNAs to the paragraphs on XPO5 and RanGTP could have smoothed the transition between these topics.

L239, "provoke actions" might be ambiguous, it could be replaced with more specific terminology like "regulate gene expression”.

In scientific writing, using the active voice can make statements more direct and assertive. L242-243 could be rephrased.

These are suggestions to enhance the readability of the manuscript and do not detract from the scientific merit of the study.

Author Response

This study aims to elucidate the association of specific XPO5 and RAN variants (c.722G>A; rs34324334 and c.*770C>T; rs14035, respectively) with susceptibility to HCC in an Egyptian cohort. The results are highly interesting and of potential significant impact. Nevertheless, several aspects of the methodology need additional detail to improve the clarity and replicability of the study.

Comment #1

  1. The manner in which genotype and allele frequencies for the XPO5rs34324334 and RANrs14035 variants were derived from the 1000 Genomes Project is not elucidated. A clearer description of this process is necessary.

Response #1

Thanks for the reviewer’s evaluation. Our team retrieved the data of genotypes and alleles of 1000 genome project for XPO5*rs34324334 and RAN*rs14035 variants from the IGSR: The International Genome Sample Resource (https://www.internationalgenome.org/data) and Ensembl database:

  https://www.ensembl.org/Homo_sapiens/Variation/Population?db=core;r=6:43566781-43567781;v=rs34324334;vdb=variation;vf=174239583

https://www.ensembl.org/Homo_sapiens/Variation/Population?db=core;r=12:130876196-130877196;v=rs14035;vdb=variation;vf=728770087

In addition, we added the sources of these databases to the manuscript as requested by the reviewer.

 Comment #2

  1. The methods utilized to perform the PCA on HCC patients carrying the XPO5rs34324334 and RANrs14035 variants require further explanation. Details about the variables included in the PCA, any data preprocessing steps undertaken, and the software or software package employed for the PCA would be beneficial.

Response #2

Thanks for the reviewer comment. We performed the Principal component analysis (PCA) based on R software version 4.2.2 with R studio version 2023.03 Build 386. The packages used in this analysis were FactoMineR” and “Factoextra”, respectively. In addition, we added this part in the statistical section.

 “Principal component analysis (PCA) categorized HCC patients into three clusters based on the genotype variants. Axis one and two for these tests accounted for 6.7% and 10.2% for the XPO5 genotypes and 7.1% and 10.2% for the RAN genotypes. This analysis showed no distinct demarcation among the patients and the XPO5 and RAN genotypes.”

In addition, we added the codes that are used in performing PCA for review.

 ###XPO5*rs34324334

library(FactoMineR, Factoextra)

fviz_pca_ind(HCCpca, axes = c(2,4),

             geom = "point", pointsize = 1.3,

             habillage = as.factor(gene3$XPO5),

             ellipses.level=0.80,

             addEllipses = TRUE,

             palette = c("#e06666", "#a500bb", "#1d390f"),

             legend.title = "XPO5 (rs34324334)") +

  scale_shape_manual(values=c(15,16,17)) + theme_prism(base_size = 10,

                                                       base_line_size = 0.7) +

  theme(legend.title = element_text("XPO5 (rs34324334)"),

        legend.text = element_text(face = "bold"),

        legend.position = "bottom")

ggsave("rami_90.png", width = 10, height = 8, units = 'in')

dev.off()

#### RanGTP*rs14035

fviz_pca_ind(HCCpca, axes = c(2,3),

             geom = "point", pointsize = 1.3,

             habillage = as.factor(gene3$RAN),

             ellipses.level=0.80,

             addEllipses = TRUE,

             palette = c("#e06666", "#a500bb", "#1d390f"),

             legend.title = "RAN (rs14035)") +

  scale_shape_manual(values=c(15,16,17)) + theme_prism(base_size = 10,

                                                       base_line_size = 0.7) +

  theme(legend.title = element_text("RAN (rs14035)"),

        legend.text = element_text(face = "bold"),

        legend.position = "bottom")

ggsave("rami_91.png", width = 10, height = 8, units = 'in')

dev.off()

 Comment #3

  1. More thorough details concerning the steps for the bioinformatics analysis of the XPO5 and RAN genes are needed in the method part. This should include information on the identification of genes, the analysis of protein-protein and gene-gene interactions, the determination of subcellular protein localization, and the data collection methods for survival analysis.

Response #3

Thanks for the reviewer. We added methods concerning “bioinformatic analysis”.

Various online databases were explored for the identification of the functional roles of XPO5 and RAN along with their potential proteins involving Ensembl database for chromosomal screening (https://www.ensembl.org/), National Center for Biotechnology Information (https://www.ncbi.nlm.nih.gov/gene/), Compartments for localization for specific proteins inside the cell (https://www.proteinatlas.org/), Kaplan‐Meier plotter database for survival analysis (https://kmplot.com/analysis/), protein data bank for the crystal structure of the specific proteins (https://www.rcsb.org/), String database for protein-protein interactions and gene ontology (https://string-db.org/), and  GeneMania database for gene prediction (https://genemania.org/). All these electronic databases were accessed on 28 February 2023.

 Comment #4

Besides that, Line 79: while the overall goal of the study is mentioned, it would be beneficial to articulate the specific purpose or hypothesis of the study more directly.

Response #4

Thanks for the reviewer hint, we modified the aim part as requested.

 Comment #5

L164-166, regarding the PCA (Figure 1E & 1F) with "no distinct demarcation among different genotypes" might need further explanation or clarification. Does this mean that these genetic variants do not stratify the patients into different groups?

Response #5

Thanks to the reviewer’s comment, we discussed this part in response #2 and explained that the analysis was performed among HCC patients only rather than cancer-free controls.

 Comment #6

Comments on the Quality of English Language

The manuscript is generally understandable, but there are several points that could improve its quality.

For example, a sentence in the introduction linking the paragraph on miRNAs to the paragraphs on XPO5 and RanGTP could have smoothed the transition between these topics.

Response #6

Thanks to the reviewer’s comment, we modified the introduction as requested.

 Comment #7

L239, "provoke actions" might be ambiguous, it could be replaced with more specific terminology like "regulate gene expression”.

Response #7

Thanks for the reviewer’s evaluation. We made the required changes.

 Comment #8

In scientific writing, using the active voice can make statements more direct and assertive. L242-243 could be rephrased.

Response #8

Thanks for your efforts, we modified and rephrase the sentence.

 Comment #9

These are suggestions to enhance the readability of the manuscript and do not detract from the scientific merit of the study.

Response #9

All the authors would send their appreciation to the respected reviewer for the time he spent in the reviewing this piece of work.

Round 2

Reviewer 1 Report

The explaination on "why Egyptian subjects were used only" in authors' response could be added in manuscript as well on a suitable place to make it understood for readers as well not only for reviewers. 

Author Response

Comment #1

Comments and Suggestions for Authors

The explanation on "why Egyptian subjects were used only" in authors' response could be added in manuscript as well on a suitable place to make it understood for readers as well not only for reviewers.

 Response #1

Thanks for the time and effort of the respected reviewer. We updated the issue within the aim of the work section. Please accept our appreciation for your precise evaluation of this manuscript.

 “Egypt has the highest incidence of HCC in Middle East with nearly ~45.9/100,000 within males and ~22.7/100,000 within females. Additionally, HCC among Egyptians is considered the second most prevalent carcinoma after breast cancer [33,34]. The present work was designed to clarify the relationship of XPO5 (c.722G>A; rs34324334) and RAN (c.*770C>T; rs14035) variants with the susceptibility to HCC among Egyptian subjects form the Delta Region of Egypt. Additionally, we performed various bioinformatic analysis regarding the XPO5 and RAN genes to focus the light on the importance of these potential variants in the progression of carcinomas.”